# HLA-G Expression/Secretion and T-Cell Cytotoxicity in Missed Abortion in Comparison to Normal Pregnancy

**DOI:** 10.3390/ijms25052643

**Published:** 2024-02-24

**Authors:** Antonia Terzieva, Marina Alexandrova, Diana Manchorova, Sergei Slavov, Lyubomir Djerov, Tanya Dimova

**Affiliations:** 1Institute of Biology and Immunology of Reproduction “Acad. Kiril Bratanov”, Bulgarian Academy of Sciences, 1113 Sofia, Bulgaria; anterzieva@abv.bg (A.T.); marmartal@abv.bg (M.A.); diana_man4orova@abv.bg (D.M.); 2Obstetrics and Gynecology Department, Medical University, University Obstetrics and Gynecology Hospital “Maichin Dom”, 1431 Sofia, Bulgaria; sergeislavov66@gmail.com (S.S.); bubodjerov@abv.bg (L.D.)

**Keywords:** HLA-G, missed abortion, T cells, γδ T cells

## Abstract

The main role of HLA-G is to protect the semi-allogeneic embryo from immune rejection by proper interaction with its cognate receptors on the maternal immune cells. Spontaneous abortion is the most common adverse pregnancy outcome, with an incidence rate between 10% and 15%, with immunologic dysregulation being thought to play a role in some of the cases. In this study, we aimed to detect the membrane and soluble HLA-G molecule at the maternal–fetal interface (MFI) and in the serum of women experiencing missed abortion (asymptomatic early pregnancy loss) in comparison to the women experiencing normal early pregnancy. In addition, the proportion of T cells and their cytotoxic profile was evaluated. We observed no difference in the spatial expression of HLA-G at the MFI and in its serum levels between the women with missed abortions and those with normal early pregnancy. In addition, comparable numbers of peripheral blood and decidual total T and γδT cells were found. In addition, as novel data we showed that missed abortion is not associated with altered extravilous invasion into uterine blood vessels and increased cytotoxicity of γδT cells. A strong signal for HLA-G on non-migrating extravilous trophoblast in the full-term normal placental bed was detected. In conclusion, HLA-G production at the MFI or in the blood of the women could not be used as a marker for normal pregnancy or missed abortions.

## 1. Introduction

For more than a half a century new, the acceptance and development of a semi-allogeneic fetus has been an enigma for scientists. Pregnancy is not just a case of acceptance or rejection but a proper and balanced interaction between the mother and the fetus, especially from an immunological point of view. The human hemochorial placenta is the most invasive type of placenta, in which there is intimate contact between the fetal trophoblast and the maternal decidua. The contact between the tissues can be referred to as the maternal–fetal interface (MFI) and can be divided into two interfaces: the first between the chorionic villi and the maternal peripheral blood, and the second between the extravilous trophoblasts (EVTs) and the decidual cells [1]. The infiltration of the MFI with certain immune cell populations is essential for pregnancy. Their presence is a key element in the regulation of trophoblast invasion, spiral artery remodeling, immune tolerance, maintenance of uterine integrity, and pathogen protection [2,3,4]. While decidual natural killer (dNK) cells represent the predominant cell population at the MFI in early human pregnancy (more than 70% of the decidual lymphocytes), T cells account for about 20% but their number is relatively stable until the end of the pregnancy [5,6,7,8,9,10]. Approximately 30–45% of T cells are CD4+ T cells, 45–75% are CD8+ T cells, and 6% of CD4 cells are T regulatory cells [11,12]. T cells become the primary immune cells of the decidua in the third trimester, mainly due to the decrease in dNK cells [13]. 

The human leukocyte antigen HLA-G was first discovered in 1982 and received its final name in 1990 [14,15]. HLA-G is a non-classical MHC-I molecule (HLA class Ib molecule) that is specifically expressed on extravilous trophoblasts (EVTs) during pregnancy [16,17]. Pathologically, HLA-G is expressed in many different types of cancer and is implicated in oncologic immune tolerance [18,19,20,21,22]. HLA-G has seven isoforms: membrane-bound (HLA-G1, HLA-G2, HLA-G3, and HLA-G4) and soluble (sHLA-G5, sHLA-G6, and HLA-G7) [23,24,25,26]. The soluble form of HLA-G (sHLA-G) is found in the culture medium of IVF embryos [27,28], maternal blood [29,30,31], amniotic fluids [32,33], and cord blood [34,35]. There are indications that soluble HLA-G is also released by cancer cells and, consequently, is detectable in the circulation of cancer patients [36,37]. The main role of HLA-G molecules during pregnancy is to protect the semi-allogeneic fetus from the maternal immune system [17,38]. HLA-G is expressed by invading EVTs and is used as a potential ligand for direct binding to the immune cell surface receptors of NK cells, T cells, B cells, and antigen-presenting cells (APCs) such as macrophages and dendritic cells residing in the maternal decidua. These receptors include ILT-2 (LILRB1), ILT-4 (LILRB2), CD8, and KIR2DL4 (CD158d). HLA-G interaction with NK cells is related to their inactivation, an important mechanism required to maintain immune tolerance at the maternal–fetal interface [39,40,41,42]. HLA-G is also the main protective molecule against the cytotoxic effects of T cells by interacting with KIR2DL4, ILT2, and ILT4. The soluble HLA-G is secreted to down-regulate CD4+ T cell proliferation, as well as to induce the apoptosis of activated CD8+ T cells [43,44]. In addition, HLA-G+ EVTs are involved in the process of placental anchoring and in the remodeling of spiral arteries. Failure of these processes is related to spontaneous abortions, preterm birth, and preeclampsia [38,45,46,47].

Spontaneous abortion is the most common adverse pregnancy outcome, with an incidence rate between 10% and 15% [48], and it refers to a pregnancy loss at less than 20 weeks of gestation in the absence of elective medical or surgical measures to terminate the pregnancy. For clinical purposes, spontaneous abortion is often subdivided into threatened, inevitable, incomplete, missed, septic, recurrent spontaneous, and complete abortions [49,50,51]. We focused on missed abortion, a condition characterized by fetal demise but no uterine activity to expel the products of conception [52]. The role of the immune system in MA is not clear. Data about possible biomarkers as effective predictors of MA are still scanty. In this study, we aimed to detect the membrane and soluble HLA-G molecule in the gestational tissue and serum of women experiencing missed abortion in comparison to women experiencing normal pregnancy. In addition, the proportion of T cells and their cytotoxic profile were evaluated.

## 2. Results

### 2.1. Maternal–Fetal Interface after Missed Abortion (MA)

We confirmed all cases of MA by patohistological analysis. As shown in Figure 1A–D, large blood clots are clearly visible in the decidua together with necrotic tissue surrounded by fibrin as well as numerous destroyed decidual glands closely associated with massive immune cells infiltration. In addition, we observed the presence of inflamed and irregular immature placental villi with oedema and numerous blood clots, unlike well-structured chorionic villi from normal early placenta (Figure 1E,F,H,I). Anchoring placental villi are specialized areas that attach the placenta to the decidua, a place from which the EVTs migrate and invade the decidua to participate in the remodeling of maternal spiral arteries and glands. Due to inflammation, in MA decidua, the anchoring villi with trophoblast columns were not clearly distinguishable compared to ones in normal pregnancy (Figure 1B,H). When we explanted the aborted placental villi and cultured them in vitro for 24 h, we did not detect blood vessels as in normal placental explants (Figure 1J,K). 

### 2.2. Same Pattern of HLA-G Expression at Maternal–Fetal Interface in MA and Normal Pregnancy

During vascular remodeling, EVTs replace endothelial cells on the walls of uterine spiral arteries and destroy their *tunica media* which is replaced with acellular fibrinoid material to increase the diameter of the vessels, turning them into large irregular sinusoids supplying the intervillous spaces with proper blood flow for adequate fetal nutrition. We detected strong HLA-G expression by EVTs invading the decidua (interstitial EVTs, iEVT) and remodeling the blood vessels (endovascular EVTs, eEVT) in both MA and normal pregnancy (Figure 2A,B,E,F). Moreover, HLA-G+ EVT plugs were found in the blood vessels (Figure 2C). eEVTs initially form plugs in the spiral arteries that limit high-pressure maternal blood flow into the intervillous space before 8–10 gw, prior to the establishment of the hemochorial circulation [53]. The invasion of HLA-G-positive EVTs in the glands was not observed either in MA or in normal decidua (Figure 2B,F). HLA-G-positive immune cells and a weak HLA-G signal on decidual stromal cells (DSCs) were detected in both normal and MA decidua (Figure 2A,B). In both cases, namely MA and normal pregnancy, the floating placental villi showed a similar pattern of HLA-G expression—negative syncytiotrophoblast and positive underlying cytotrophoblasts (Figure 2D,H). In some samples derived from normal placenta, we detected HLA-G+ “patches” on the syntitiotrophoblasts (Figure 2H). The trophoblast columns of the clearly visible anchoring villi of the normal placenta contained intermediate cytotrophoblasts strongly positive for HLA-G, like those in MA placenta (Figure 2D,G). As we mentioned already, a massive infiltration of immune cells was noted in the MA decidua, some of which were HLA-G-positive (Figure 2B). We detected HLA-G-positive immune cells in normal decidua as well. 

### 2.3. Expression of Non-Classical HLA-G Molecule in Full-Term Placenta

As is shown in Figure 3A–C, we visualized a full-term placental bed and placental villi. Our results show that the full-term placental bed contains large clusters of EVT with a morphology of sedentary (non-migratory) cells (Figure 3A,B) with strong HLA-G expression (Figure 3D,E).

In addition, weaker expression was found in cytotrophoblast cells inside the chorionic villi, while syncytiotrophoblasts covering the villi were HLA-G-negative (Figure 3F).

### 2.4. Comparable Levels of sHLA-G in the Sera of Women with MA and with Normal Early Pregnancy

The serum levels of sHLA-G protein of MA women (n = 17) and women with normal pregnancy (control group, n = 30) were compared. No difference between both groups (87 ± 4 ng/mL vs. 91 ± 17 ng/mL, *p* = 0.4261) was found (Figure 4a). We performed another comparison by dividing the existing and adding new groups as follows: the MA group was divided in two groups: women with (n = 6) and with no previous live birth/s (n = 9), the control group of women in early normal pregnancy was the same (n = 17), and two new groups were added—full-term pregnant women (n = 6) and non-pregnant, non-laboring women (n = 8). Again, no difference in the serum levels of HLA-G was observed when all these groups were compared (Figure 4b). 

### 2.5. Similar Proportions of Total T Cells, γδ T Cells, and Cytotoxic T Cells in MA and Normal Pregnancy

Since normal early pregnancy is generally associated with the down-regulation of the adaptive immune system (conventional T-cell responses) and specific adaptations of the innate immune cell populations at the MFI, our focus was on T cells, unconventional subset γδ T cells, and their potential for cytotoxicity. Our FACS analysis of total T-cell proportion in the blood and decidua of MA women compared to those with normal early pregnancy showed no significant difference either in the numbers of decidual T cells (*p* = 0.6158) or of peripheral blood T cells (*p* = 0.4071, Figure 4c). No difference was observed in the number of peripheral blood and decidual γδT cells either (*p* = 0.8450, *p* = 0.0683, respectively, Figure 4d). To evaluate the cytotoxic potential of conventional (αβ) and non-conventional (γδ) T cells, we examined the intracellular expression of the pore-forming molecule perforin and cytotoxic granzyme B. Although showing relatively high cytotoxic potential, γδ T cells in the MA decidua did not produce more perforin and granzyme B as compared to those in normal decidua (*p* = 0.3095 and *p* = 0.1139, respectively, Figure 4e,g). The same results were found for peripheral blood γδ T cells positive for perforin or granzyme B in both groups (*p* = 0.1429 and *p* = 0.2721, respectively, Figure 4e,g). While the levels of peripheral blood perforin+αβT cells+ were comparable in both groups (Figure 4f), in the aborted decidua αβ T cells positive for perforin had a lower number (*p* = 0.0159, Figure 4f). No difference was found in the proportion of αβT cells positive for granzyme B either in the decidua (*p* = 0.1949) or in the blood (*p* = 0.8518) between women experiencing missed abortions and those experiencing normal pregnancies (Figure 4h). 

## 3. Discussion

The data of our study are derived from the completion of HLA-G expression at the direct site of maternal–fetal contact with systemic levels in both normal and aborted pregnancy. We completed this study with novel data about cytotoxicity of decidual T cells.

Our results show several important findings: (1) no difference in the spatial expression of HLA-G at the MFI in MA and normal pregnancy; (2) strong signal for HLA-G on non-migrating EVTs in the full-term normal placental bed; (3) similar serum levels of sHLA-G in both women experiencing missed abortion and normal pregnancy as well as in non-pregnant women; and (4) women experiencing aborted and normal pregnancy showed comparable numbers of peripheral blood and decidual total T cells and γδT cells as well as cytotoxic γδT cells. 

EVTs that invaded decidua strongly expressed HLA-G regardless of the experience of normal pregnancy or MA. Thus, our finding excludes the possibility that MA could be explained solely by spatial differential HLA-G expression by EVTs. Our result is consistent with previous studies showing no difference in the expression of HLA-G between women experiencing recurrent miscarriage (RM) and those experiencing normal pregnancy [54,55,56,57] but contradicts the observation of Moreau et al. regarding the lower intensity of HLA-G staining of iEVT from women experiencing RM [58]. It should be noted that, in our MA group, two of the women only experienced RM, the rest experienced a single abortion. For the first time, we showed endovascular trophoblast invasion into uterine arteries as well as plugs with HLA-G+ endovascular EVTs in blood vessels in MA cases like in normal early pregnancy. The detection of HLA-G-positive eEVT plugs in MA decidua suggests that MA is not associated with altered EVT invasion into blood vessels and/or failed vascular remodeling. Some authors described a compromised EVT invasion into the venous and lymphatic vasculature in recurrent spontaneous abortion [59]. We cannot, however, exclude different degrees of blood vessel remodeling in both groups as a possible reason of pregnancy disorder. It seems that HLA-G expression in EVTs is an intrinsic property of cell differentiation and HLA-G production is a critical component of cytotrophoblast differentiation along the invasive EVT pathway [60]. In line with our previous in vitro study, we demonstrated the rapid differentiation of primary early pregnancy cytotrophoblasts to EVTs with invasive property, concomitant HLA-G and HLA-C expression, and a hybrid (Vim+/CK7+) phenotype [61]. Some authors reported that elevated HLA-G expression found in severe preeclampsia cases may reflect undifferentiated status of EVTs [62] and that EVTs from a full-term placental bed having reduced invasive capacity also have a decreased ability to up-regulate HLA-G protein expression [63]. However, we detected a strong expression of a non-classical HLA-G molecule in EVTs, located in a full-term placental bed. Recently, the long-lasting dogma that the strong HLA-G expression is a specific feature of early pregnancy invasive and migrating EVTs was denied by the observation of Papuchova et al. according to which term pregnancy EVTs showed higher levels of HLA-G (measured by FACS) as compared to first trimester EVTs [64]. Interestingly, HLA-G expression was increased in EVTs from term placental bed in the pregnancy with a male fetus [64] as well as in the normal pregnancy of women with a history of RM [65]. Some researchers associated increased HLA-G expression in the basal plate of the placenta with preterm birth [66]. Detailed characterization of HLA-G+ EVT from term pregnancy in comparison to first trimester revealed their unique phenotypes and gene expression profiles [64]. It is well known that EVTs in a full-term placental bed are immobile cells undergoing endoduplication and become so-called placental bed giant cells. How the strong HLA-G expression is related to the stationary nature of the EVTs in the full-term placental bed remains to be elucidated. The finding that HLA-G was expressed in EVTs from ectopic pregnancy suggests that this expression is induced in a cell-autonomous manner rather than determined by appropriate environmental cues [60]. There is still a controversy about which kinds of cells at the MFI, other than EVTs, express HLA-G isoforms and it seems that this depends on the antibodies and immunohistochemical methods used [67]. In addition to strong HLA-G immunostaining by iEVT and eEVT in the decidual stroma (normal and MA), we confirmed a weak signal on some DSC in accordance with previous data showing that this expression on DSC could be modulated by cytokines and decidualization [68]. We also noted HLA-G-positive immune cells at the MFI in both aborted and normal tissues. It has been shown that the establishment of the immune tolerance involved the interaction of EVTs with dNK cells, in which dNK cells acquire HLA-G from EVTs by trogocytosis [39]. Named after the Greek word trogo (to nibble), trogocytosis consists in membrane protein transfer between cells. Regarding the pattern of HLA-G expression in placental villi, we found that both groups—MA and normal pregnancy—have a similar pattern of HLA-G staining, i.e., HLA-G-negative syncytiotrophoblasts (floating villi) and HLA-G-positive intermediate cytotrophoblasts in the column of anchoring villi. In line with this, Patel et al. observed that women with idiopathic recurrent pregnancy loss exhibited an expression of HLA-G in the anchoring villi, whereas no such expression was detected in floating villi [54]. In some normal pregnancy samples, we detected “patched” staining for HLA-G on syntitiotrophoblast, which could be explained by the presence of the HLA-G1 isoform [69]. Although thoroughly studied, the presence and role of sHLA-G in the circulation of pregnant women is still not clear. sHLA-G is likely to be released by endovascular trophoblasts invading maternal spiral arteries and therefore to be present in the maternal blood circulation. Several authors hypothesized that such soluble forms may act as specific immunosuppressors during pregnancy [70]. The sHLA-G protein has been detected in serum from pregnant women, non-pregnant women and normal males, amniotic fluid, and cord blood [71,72]. Similarly, we detected sHLA-G in the serum of non-pregnant women, and we did not find a distinction in serum sHLA-G levels between women experiencing MA and experiencing normal pregnancy. In accordance with our findings, other studies showed comparable sHLA-G levels in women experiencing normal pregnancy and women having aborted before the 25th week of gestation [73,74]. Thus, sHLA-G may not prove to be a functional early non-invasive biomarker of normal pregnancy or missed abortion. Moreover, it has been shown that a high level of HLA-G may not be necessary for a healthy pregnancy [73]. By following the expression of certain sHLA-G isoforms, Steinborn et al. reported that a measurement of sHLA-G1/G5 plasma levels may be a powerful new tool in prenatal diagnostics for the identification of women with an increased risk of intrauterine growth retardation and/or preeclampsia [73]. Since T cells have always been considered the critical cells for immune balance regulation, many studies have investigated the role of T cells at the MFI over the past three decades (rev in [75]). However, the specific features of T cells have not been fully elucidated. Normal pregnancy or RSA is rather associated with a complex dynamic of Th, Tc, and Treg cells and their subsets at the MFI than a dramatic change in the T-cell total number [76,77]. Indeed, we did not find a difference in total T-cell and γδT-cell numbers in the blood and decidua between MA women and those experiencing normal pregnancy. TCRγδ T cells represent a minor part of T cells but they are enriched at the MFI during early human pregnancy, showing decidua-specific phenotypes [78,79]. A predominance of activated and terminally differentiated pro-inflammatory and cytotoxic γδ T-cell effectors at the MFI was observed [79]. The expression level of cytotoxic granules is important to physiological pregnancy; however, the abnormal expression of cytotoxic granules by NK cells may contribute to adverse pregnancy events, including spontaneous abortion [80,81]. In addition to NK cells, γδ-T cells can also exert cytotoxic activity in many physiological and pathological processes through cytotoxic granules and share several receptors with NK cells, including activating and inhibitory NK receptors [82,83,84,85]. In vivo, Huang et al. monitored the expression of perforin and granzyme B in the blood of women with unexplained repeated implantation failure (uRIF) after IVF-ET. They found that the increased proportion of peripheral GrB+ γδ-T cells in lymphocytes was significantly associated with clinical pregnancy failure in patients with uRIF [86]. While decidual γδ T cells expressed less perforin than their counterparts in the blood, they expressed significantly more granulysin during early pregnancy. Strikingly, this high granulysin expression was limited to early pregnancy, as it was reduced at term pregnancy [87]. In this study, as novel data, we examined and found the same levels of cytotoxic γδ or αβ T cells producing granzyme B in both MA deciduas and in normal ones. The same is valid for the peripheral blood as well. Therefore, the cytotoxicity of T cells could not be blamed as a reason or consequence of the missed abortion. Single abortions as a normal part of the highly ineffective human reproduction process may not be accompanied with a significant difference in HLA-G levels and in the numbers of granzyme B+ cytotoxic T cells. It seems that, although present at the MFI, the fine-tuning of the intensity of the HLA-G signal could be important for an adequate interaction with KIRs on the resident immune cells in normal pregnancy. Moreover, we do not know how many abortion cases are due to chromosomal abnormalities which are the most prominent cause for spontaneous pregnancy loss. Inclusion of such MA cases in the current study could mask the real role of the immune factors.

In conclusion, there is no differential expression of HLA-G at the MFI and different serum levels of sHLA-G in the blood of the women experiencing missed abortion as compared to women experiencing normal early pregnancy. The numbers of peripheral blood and decidual T cells, γδT cells, and cytotoxic γδT cells are comparable in the women experiencing missed abortion and experiencing normal pregnancy. The main source of HLA-G at the MFI from the beginning to the end of the pregnancy is EVTs, which as stationary giant cells in the full-term placental bed strongly express HLA-G. 

## 4. Materials and Methods

### 4.1. Study Populations and Samples

Healthy pregnant women experiencing missed abortions (MAs, 6–12 gw, n = 25), experiencing normal early pregnancy, directed to elective pregnancy termination (6–15 gw, n = 20), in term pregnancy, directed to delivery (38–40 gw, n = 5), as well as healthy non-pregnant women in reproductive age (volunteers, n = 16) were involved in this study. The gestational sacs of all women experiencing MAs were confirmed at 7–11 weeks in gestational age under ultrasonography. First-trimester missed abortion was defined as an intact gestational sac lacking any fetal cardiac activity, intrauterine gestational sac with the largest diameter exceeding 10 mm but devoid of yolk sac, or an empty gestational sac with a confirmed gestational age of no less than 7 weeks [88]. First-trimester decidual tissues from elective terminations of pregnancies were examined and used as a reference for healthy pregnancy. This study was carried out in accordance with the Declaration of Helsinki and was approved by Human Research Ethics Committee at the University Obstetrics and Gynecology Hospital “Maichin Dom” and the Medical University, Sofia, Bulgaria (No 250569/2018). The women were notified of the study and signed an informed consent document. Gestational tissues (decidual and placental) obtained after missed or elective abortions as well as after normal term delivery were assessed by histology and subjected to immunohistochemical detection of HLA-G. Blood samples from women in early pregnancy and after MA, and from non-pregnant ones, were used for detection of sHLA-G. Paired samples blood and decidua from women experiencing MA as well as from those in normal early pregnancies were analyzed by FACS for T cell proportions. The samples were processed within one hour after blood withdrawal and tissue collection.

### 4.2. Histological Examination and Immunohistochemistry (IHC)

For histological examination, the gestational tissue pieces (10 mm × 10 mm × 10 mm) were fixed in HOPE^®^ (Polysciences, EuropeGmH, Hirschberg an der Bergstrasse, Germany), routinely processed, embedded in paraffin wax, and sectioned at 7 μm. The sections were stained with hematoxylin and eosin (H&E). Selected slides were subjected to IHC for in situ detection of HLA-G. A three-step biotin–streptavidin–enzyme method and Biotin-HRP anti-rabbit visualization system (ScyTek, Logan, UT, USA) were used. Briefly, dewaxed and rehydrated sections were incubated with Super Block to inhibit the non-specific binding. Endogenous peroxidase activity was blocked with 3% H_2_O_2_ for 30 min at 37 °C. Then, the sections were incubated with primary polyclonal rabbit anti-human antibody HLA-G (E-AB-18031, Elabscience, Houston, TX, USA) overnight at 4 °C in a humidified chamber. After washing the slides with PBS thrice, the endogenous biotin was blocked and then the sections were incubated with biotinylated antibody for 10 min at room temperature (RT). The slides were washed thrice and incubated with streptavidin-horseradish peroxidase for 10 min at RT. The visualization was carried out with chromogen—DAB (3,3 diaminobenzidine tetrahydrochloride). Nuclei were counterstained with hematoxylin. For negative control, the primary antibody was omitted. Sections from human breast cancer tissue processed in the same way were used as positive controls for specificity of HLA-G staining (Figure 2I).

### 4.3. ELISA

Blood samples were collected in serum collection tubes (KABE Laboratortechnik, Wiehl-Bielstein, Germany) and incubated at 37 °C for 30 min for a clot to form, then centrifuged for 10 min at 3000 rpm. The sera were collected, aliquoted, and stored at −80 °C. The measurement of sHLA-G protein in serum samples was performed according to the manufacturer’s instructions (Human MHCG/HLA-G ELISA kit, Elabscience®, Houston, TX, USA). Serum samples (diluted 1:2) were run in duplicate alongside serial dilutions of human HLA-G standard. The color density related to the concentration of sHLA-G protein in the serum samples was detected at 450 nm using an LKB 5060-006 ELISA Reader (LKB Instruments, Vienna, Austria). The serum levels of sHLA-G were calculated according to the standard curve. The sensitivity of the ELISA test was 0.38 ng/mL, and the detection range was 0.63−40 ng/mL. The concentration of HLA-G was calculated according to the manufacturer’s instructions, using the software CurveExpert 1.4.

### 4.4. Mononuclear Cells Isolation (PBMC, DMC)

Blood samples were obtained in heparin anti-coagulated vacutainer tubes (BD Biosciences, San Jose, CA, USA). Peripheral blood mononuclear cells (PBMCs) were isolated from blood samples diluted with PBS (1:2) by Lymphoprep density gradient centrifugation method (20 min/800× *g*, density: 1.077 g/mL, Sigma-Aldrich, Munich, Germany). The aliquots of PBMCs were used immediately for FACS staining. Decidual tissue (normal and spontaneously aborted, first trimester) was carefully separated from trophoblasts. Third-trimester decidua basalis (term placenta) was dissected from the maternal-facing surface of the basal plate, covered by decidua basalis. To avoid selective cell death or selective loss of surface proteins, mechanical disintegration rather than enzymatic digestion was used to process the decidual tissue and to isolate the decidual leukocytes. Separation of decidual mononuclear cells (DMCs) from early and term decidual tissues were prepared by mechanical disruption of tissue in sterile PBS (5 g/50 mL) followed by sequential filtrations of resultant suspension with a 100 μm metal sieve and a 60 μm strainer (Becton Dickinson, San Jose, CA, USA), and centrifugation at 1500 rpm for 15 min. The pellet was resuspended in sterile PBS, layered on Lymphoprep and spun at 800× *g* for 20 min (without break). The mononuclear cells were removed from the interface, washed, assessed for viability with trypan blue exclusion (always achieving a purity greater than 95%), and then were used for FACS staining. The yield was usually 0.5–1 × 10^6^ cells per gram tissue. 

### 4.5. Flow Cytometry

Freshly separated untouched PBMCs and DMCs adjusted to 1 × 10^6^ cells per sample were used for surface staining of T cell markers and for intracellular staining of cytotoxic molecules perforin and granzyme B. For subset identification, the suspensions were incubated with the following monoclonal antibodies (mAbs) in different combinations: CD3–FITC (clone UCHT1, ImmunoTools, Friesoythe, Germany), CD3 APC (clone UCHT-1, BD Biosciences, San Jose, CA, USA), γδ TCR-PE (clone F11; BD Biosciences), perforin-FITC (clone dG9, BD Biosciences), granzyme B-FITC (clone GB11, BD Biosciences), isotype IgG2-FITC, and IgG1-FITC (both from BD Biosciences). The cells were washed with 3 mL ice-cold FACS buffer (PBS containing 0.1% bovine serum albumin, Sigma-Aldrich, Munich, Germany) and incubated with the mAbs for 20 min at 4 °C in the dark. After washing with FACS buffer, the cells were fixed in 300 μL 1% paraformaldehyde (Sigma-Aldrich). For intracellular staining (granzyme A, perforin), the Cytofix/Cytoperm kit (eBioscience, San Diego, CA, USA) was used. Flow cytometric analyses were performed on an FACS Calibur and data were processed using FloJo software (Treestar, San Carlos, CA, USA). A real-time gate was set around the viable lymphocytes based on their forward scatter/side scatter profile. Approximately 50,000 cells per sample were acquired for analysis. Compensation controls were prepared simultaneously with sample processing using cells stained with a single mAb. Fluorescence minus one (FMO) and isotype-matched immunoglobulins were used as controls for nonspecific immunofluorescence and to set gates. T lymphocytes were selected based on forward and side scatter plots and staining for CD3. Within the CD3 + cell population, γδ T cells were distinguished by staining with anti-γδTCR mAb. By gating on γδT cells and non-γδT cells (αβT cells), the proportions of perforin- and granzymeB-positive cells were analyzed. 

### 4.6. Statistical Analyses

Statistical analyses were performed using GraphPad Prism 8.0.1 software (GraphPad Software Inc., La Jolla, CA, USA). The Mann–Whitney test was performed to test for significant differences in serum levels of HLA-G between the different groups of women. For comparisons of independent groups, a Student *t*-test or the Mann–Whitney test was performed. For comparisons of matched groups, a paired Student *t*-test or Wilcoxon matched test was performed. For all results, a *p*-value of <0.05 was considered statistically significant.

## Figures and Tables

**Figure 1 ijms-25-02643-f001:**
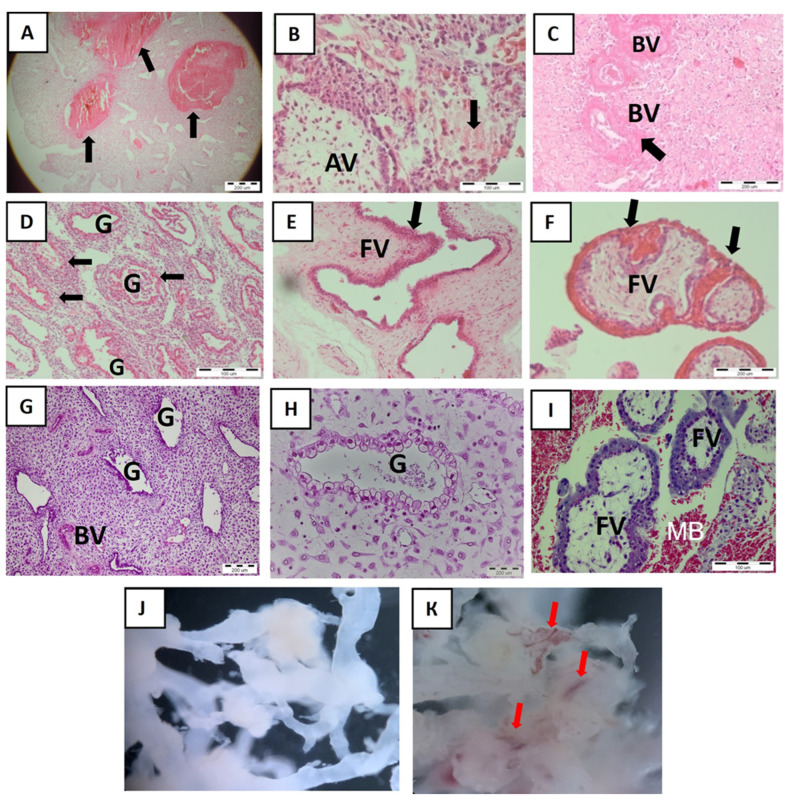
Maternal–fetal interface and explanted placental villi in MA (**A**–**F**,**J**) and normal early pregnancy (**G**–**I**,**K**). (**A**–**F**): MA decidua with massive blood cloths (black arrow, **A**) and anchoring villa and blood vessels with necrotic tissue (black arrow, (**B**) and (**C**), respectively). Note the destroyed decidual glands with massive immune infiltration around (**D**). The placental villi are immature, inflamed, and irregular in shapes (black arrow, **E**), showing numerous clots (black arrow, **F**). Representative images for n = 10. Explanted placental villi derived from aborted pregnancy (**J**, n = 3) are not vital, and they are shown by a lack of blood vessels which are numerous in explanted normal placental villi (red arrows, **K**). (**G**–**I**): Healthy decidua with normal glands and blood vessels (**G**,**H**) as well as floating mature placental villi bathed in the maternal blood into intervillous spaces (**I**). Representative images for n = 10. Hematoxylin and eosin staining. BVs—blood vessels, G—gland, AV—anchoring villa, FV—floating villi, MB—maternal blood. Scale bar represents 200 μm (**A**,**C**,**F**,**G**,**H**) and 100 μm (**B**,**D**,**I**). Magnification 4× (**J**,**K**) and 10× (**E**).

**Figure 2 ijms-25-02643-f002:**
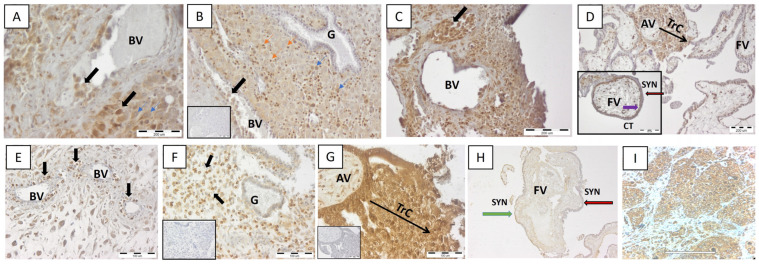
In situ expression of non-classical HLA-G molecule at MFI in MA (**A**–**D**) and normal early pregnancy (**E**–**H**). Representative images of n = 10 for each group. Note the similar pattern in HLA-G expression in decidua and placental villi (anchoring and floating) between women with MA and normal pregnancy. HLA-G+ EVT invading the decidua and blood vessels (black arrow, **A**,**B**,**E**) but not the decidual glands (**B**,**F**). Endovascular plugs of HLA-G+ EVT in aborted decidua (black arrow, **C**). HLA-G+ immune cells (orange arrow) as well as some weakly positive signals for HLA-G DSC (blue arrows) were observed (**A**,**B**). Trophoblast columns (TrC) of the anchoring villa contain intermediate cytotrophoblasts positive for HLA-G (**D**,**G**). The syntitiotrophoblast of the floating villi do not express HLA-G (red arrow, **D**,**H**) unlike underlying cytotrophoblasts (purple arrow, **D**). (**I**)—HLA-G-positive breast cancer tissue serving as positive control for the specificity of the staining. Negative control sections are given as insets in (**B**,**F**,**G**), and the inset in (**D**) shows floating villa at higher magnification. BVs—blood vessels, G—gland, AV—anchoring villa, FV—floating villi, syn—syntitiotrophoblasts, CTs—cytotrophoblasts, TrC—trophoblast column. Scale bar represents 200 μm (**A**,**C**,**D**), 180 μm (**I**), and 100 μm (**E**,**F**,**G**). Magnification 10× (**B**) and 20× (**H**). The scale bars of the insets correspond to those of the main images, except for (**D**) where the inset scale bar is 100 μm.

**Figure 3 ijms-25-02643-f003:**
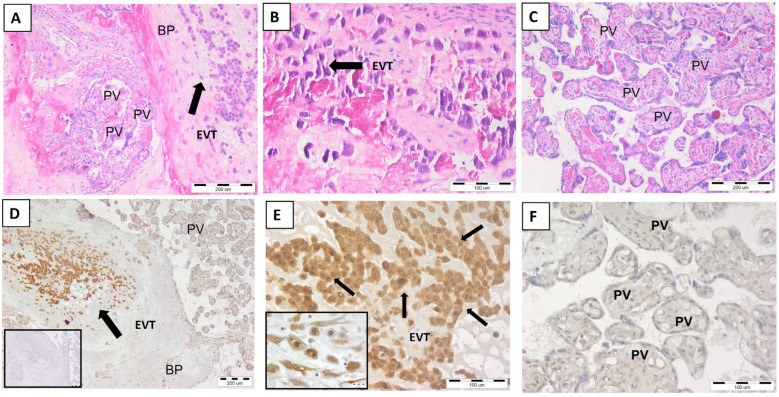
The presence of EVT cells in the basal plate of full-term placenta. (**A**)—full-term placenta and EVT cells (black arrow) in its basal plate; (**B**)—EVT at higher magnification; (**C**)—placental villi; (**D**)—HLA-G-positive EVT in the basal plate of the placenta (black arrow); (**E**)—HLA-G+ EVT cells (black arrows) at higher magnification; the inset shows round-shape stationary EVTs with endoduplication; (**F**)—HLA-G-negative placental villi. (**A**–**C**)—hematoxylin and eosin staining. (**D**–**F**)—detection of HLA-G; an inset of (**D**) is a negative control staining. PV—placental villi, BP—basal plate, EVTs—extravillous trophoblasts. Representative images of n = 5. Scale bar represents 200 μm (**A**,**C**,**D**) and 100 μm (**B**,**E**,**F**). The scale bar of the inset of (**D**) correspond to that of the main image, the scale bar of the inset of (**E**) is 20 μm.

**Figure 4 ijms-25-02643-f004:**
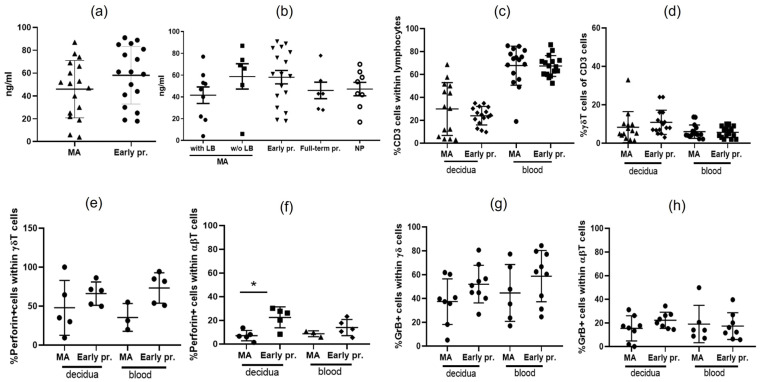
Serum sHLA-G levels, T-cell, and cytotoxic T-cell proportions of women with MA and normal pregnancy. No significant difference in the serum levels of sHLA-G was detected between women experiencing MA and those experiencing normal pregnancy (**a**) as well as between women experiencing MA with or without live birth, with normal early or late pregnancy and non-pregnant, non-laboring women (**b**); (**c**) T-cell number in the blood and decidua of women experiencing MA and normal pregnancy; (**d**) γδT-cell number in the blood and decidua of women experiencing MA and normal pregnancy; (**e**) γδT cells positive for perforin in the blood and decidua of women experiencing MA and normal pregnancy; (**f**) αβT cells positive for perforin in the blood and decidua of women experiencing MA and normal pregnancy; (**g**) γδT cells positive for granzyme B in the blood and decidua of women experiencing MA and normal pregnancy; (**h**) αβT cells positive for granzyme B in the blood and decidua of women experiencing MA and normal pregnancy. MA—missed abortion, LB—live birth, NP—non-pregnant. Mann–Whitney test, GraphPad Prism v8, * *p* < 0.05.

## Data Availability

The data presented in this study are available on request from the corresponding author.

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
