# Peer review of "HLA-G Expression/Secretion and T-Cell Cytotoxicity in Missed Abortion in Comparison to Normal Pregnancy"

_ijms, 2024, doi:10.3390/ijms25052643_

Round 1

Reviewer 1 Report

Comments and Suggestions for Authors

The manuscript is fascinating and well-written. The whole body of the manuscript and the research are designed appropriately. The discussion is very interesting. The data presented in the manuscript are very important for studies on RSA.

Author Response

We thank the reviewer for the positive comments!

Reviewer 2 Report

Comments and Suggestions for Authors

The submitted paper is a nice research on the role of HLA-G in miscarriages. The paper is overall well written and I just have few minor comments:

- few typos are presents (e.g. villi instead of villa) 

- In Figure 2 there is no explanation for arrows TrC in both panels G and D.

- In Figure 3 panels D and E there is no reference to the black arrows. Regarding E what does incl inset mean?

Author Response

Comments and Suggestions for Authors

The submitted paper is a nice research on the role of HLA-G in miscarriages. The paper is overall well written and I just have few minor comments:

We thank the reviewer for the positive comments! We answered point-by-point to his/her notes. Our redactions and additions are given in yellow in the revised version of the manuscript.

- few typos are presents (e.g. villi instead of villa)

We did a correction (Lines 108, 144, 149)

- In Figure 2 there is no explanation for arrows TrC in both panels G and D.

We added an explanation for TrC in the legend of Figure 2 (line 144, line 149).

- In Figure 3 panels D and E there is no reference to the black arrows. Regarding E what does incl inset mean?

We added a reference to the black arrows in the legend of Figure 3, panels D and E (lines 162-164). We explained as well what the inset of Figure 3, panel E does mean – Line 164.

Reviewer 3 Report

Comments and Suggestions for Authors

a very important scientific question that was well developed, clear presentation of the results

Author Response

Comments and Suggestions for Authors

a very important scientific question that was well developed, clear presentation of the results

We thank reviewer for the positive comments!

Reviewer 4 Report

Comments and Suggestions for Authors

This study examines and evaluates the expression of membrane HLA-G and serum soluble HLA-G molecules in pregnant women with missed abortions, as well as the percentage of T cells and their cytotoxicity profiles, compared to pregnant women with normal pregnancies who opt for early termination. The authors report that there were no significant differences between normal pregnancies and pregnant women with missing abortions.

Major point

There are already many papers on HLA and miscarriage, and there are even reviews and meta-analyses. I do not understand the novelty of this paper.

The abstract ends with a list of four results and no conclusion.

Figures 1, 2, and 3 are only comparisons of pathology results with N of 1 each, and are not quantitative, so this cannot be considered a comparison between two groups.

Abstract and Discussion have exactly the same text, and Abstract should be simplified.

I would like to suggest to the authors that the causes of missed abortion should be considered. Chromosomal abnormalities are the primary cause of comorbid miscarriages, which account for about 50% of cases, and immunological factors are much less likely to be the direct cause. It is possible that the inclusion of miscarriages caused by chromosomal abnormalities masked the cases of missed abortion due to immunological factors, resulting in a non-significant difference.

Minor point

The abbreviation for MA is explained in Line 335, and until then it is listed as MA with no explanation.

The scale bar on the pathology photo is not attached, or even if it is attached, the letters are too small to read.

Author Response

Comments and Suggestions for Authors

This study examines and evaluates the expression of membrane HLA-G and serum soluble HLA-G molecules in pregnant women with missed abortions, as well as the percentage of T cells and their cytotoxicity profiles, compared to pregnant women with normal pregnancies who opt for early termination. The authors report that there were no significant differences between normal pregnancies and pregnant women with missing abortions.

We appreciate very much the comments of the reviewer that allowed us to improve our manuscript. We answered point-by-point to all comments of the reviewer. Our corrections and additions are given in yellow in the revised version of the manuscript.

Major point

・There are already many papers on HLA and miscarriage, and there are even reviews and meta-analyses. I do not understand the novelty of this paper.

We agree with the reviewer’s note. Although plenty of papers on HLA-G production and abortion still a lot of controversy exists which motivate new studies in that area to accumulate new data.

The novelty of our HLA-G and missed abortions study is the complex approach of investigation of this molecule – at local and systemic level. Also, in this study we report novel data about the cytotoxicity of the decidual T cells. For the first time we showed endovascular trophoblast invasion into uterine arteries (Fig. 2 A) and plugs with HLA-G+ endovascular EVT in blood vessels (Fig. 2 C) in MA cases like in normal early pregnancy pointing out the presence of vascular remodeling.

To underline the novelty of the study we added in the Discussion several sentences as follows:

Line 221: “We completed this study with novel data about cytotoxicity of decidual T cells” (Line 221)

Lines 238-240: “For the first time we showed endovascular trophoblast invasion into uterine arteries as well as plugs with HLA-G+ endovascular EVT in blood vessels in MA cases like in normal early pregnancy.

 …. as novel data, we examined and found ….. (Line 326)

・The abstract ends with a list of four results and no conclusion.

We added Conclusion sentence to the abstract – Line 23.

・Figures 1, 2, and 3 are only comparisons of pathology results with N of 1 each, and are not quantitative, so this cannot be considered a comparison between two groups.

We clarified in the legend of Figures 1, 2, 3 that the shown images are representative of certain number of the evaluated samples in each group – Line 104, 107, 139, 167.

・Abstract and Discussion have exactly the same text, and Abstract should be simplified.

We considered the note of the reviewer and modified and simplified the Abstract – Lines 19-22.

・I would like to suggest to the authors that the causes of missed abortion should be considered. Chromosomal abnormalities are the primary cause of comorbid miscarriages, which account for about 50% of cases, and immunological factors are much less likely to be the direct cause. It is possible that the inclusion of miscarriages caused by chromosomal abnormalities masked the cases of missed abortion due to immunological factors, resulting in a non-significant difference.

We have no information about the chromosomal abnormalities of the embryos of the women with missed abortions. We thank reviewer for his/her suggestion to consider this main cause for the spontaneous abortions. Thus, we included in the end of the Discussion the next sentence, Lines 342-345:

 “Moreover, we do not know how many of abortion cases are due to chromosomal abnormalities which are the most prominent cause for spontaneous pregnancy loss. Inclusion of such MA cases in the current study could mask the real role of the immune factors.”

Minor point

・The abbreviation for MA is explained in Line 335, and until then it is listed as MA with no explanation.

We explained the abbreviation for MA at the place of its first use in the first subtitle of Results section – Line 83.

・The scale bar on the pathology photo is not attached, or even if it is attached, the letters are too small to read.

The scale bar values are additionally added in the legend of Figures 1,2 and 3 – Line 109, 110, 150, 167.

Reviewer 5 Report

Comments and Suggestions for Authors

1.      I recommend that the HLA-G-positive immune cells of the aborted and normal decidua be presented in Figure 2 (line 134)!

2.      The men included in the study are not included in the Materials and Methods. Fill in the missing data! Were there men in the ethical clearance (No 250569/2018)? If not, omit the data (line 177) from the results! This data is irrelevant.

  3. Why was it necessary to block endogenous peroxidase in samples embedded in wax (line 363)? I find it unlikely that the peroxidase would have enzyme activity once the wax is removed.

4. In addition to the quantitative determination of HLA-G, it would have been interesting to determine the monomer-dimer distribution using the western blot method. If there is a usable serum sample left, it would be worthwhile to perform the test. Maybe there is a difference in dimerization.

5. In the conclusion, a possible explanation should be given as to why there is no significant difference in the HLA-G level of the groups.

Author Response

Comments and Suggestions for Authors

We appreciate very much the comments of the reviewer that allowed us to improve our manuscript.

We answered all the comments of the reviewer, and our corrections are given in yellow in the revised version of the manuscript.

  1. I recommend that the HLA-G-positive immune cells of the aborted and normal decidua be presented in Figure 2 (line 134)!

HLA-G positive immune cells are presented in Fig. 2 B, in MA decidua, shown by orange arrows. We detected HLA-G positive cells in normal decidua as well, but data are not shown.  We added a sentence (Line 134, 135) to point out HLA-G+ immune cells in the Fig. 2.

  1. The men included in the study are not included in the Materials and Methods. Fill in the missing data! Were there men in the ethical clearance (No 250569/2018)? If not, omit the data (line 177) from the results! This data is irrelevant.

We agree with the note of the reviewer and deleted the men data from the results.

  1. Why was it necessary to block endogenous peroxidase in samples embedded in wax (line 363)? I find it unlikely that the peroxidase would have enzyme activity once the wax is removed.

We use the blocking of endogenous peroxidase as obligatory step in our immunohistochemistry protocol. Into decidua samples there are many blood vessels and in aborted ones - numerous blood clots. Thus, plenty of erythrocytes contains endogenous peroxidase which activity could not be eliminated easily. We would like to be sure that the false signal due to endogenous peroxidase is eliminated.

  1. In addition to the quantitative determination of HLA-G, it would have been interesting to determine the monomer-dimer distribution using the western blot method. If there is a usable serum sample left, it would be worthwhile to perform the test. Maybe there is a difference in dimerization.

We thank very much for reviewer’s suggestion.

  1. In the conclusion, a possible explanation should be given as to why there is no significant difference in the HLA-G level of the groups.

One possible explanation why there is no significant difference in HLA-G level between the groups could be the fact that MA group includes just cases of single spontaneous abortions as a part of ineffective human reproduction process. Other explanation could be that a fine tuning in the intensity of HLA-G signal rather than presence/absence of HLA-G expression is related to its inadequate recognition by KIRs on the immune cells at maternal fetal interface leading to abortion.

We do not know how many of MA cases are due to chromosomal abnormalities which are the most prominent cause for spontaneous pregnancy loss. Inclusion of such MA cases in the study could mask the real role of the immune factors as HLA-G.  

In conclusion part (Lines 338-345) we gave possible explanation of our findings and added the text:

“Single abortions as normal part of highly ineffective human reproduction process may not be accompanied with significant difference in HLA-G levels and in the num-bers of granzyme B+ cytotoxic T cells. Seems that although present at MFI the fine tuning of the intensity of HLA-G signal could be important for adequate interaction with KIRs on the resident immune cells in normal pregnancy. Moreover, we do not know how many of abortion cases are due to chromosomal abnormalities which are the most prominent cause for spontaneous pregnancy loss. Inclusion of such MA cases in the current study could mask the real role of the immune factors.”  

Round 2

Reviewer 4 Report

Comments and Suggestions for Authors

You are correcting the part you commented on.

However, I do not understand the importance and significance of this study. What is the medical significance of these results? Will they help in the future development of medicine? Can you reduce miscarriages from this study? I don't think the authors have explained the significance.

Author Response

We thank the reviewer for this question. As we mentioned in Introduction according to the clinicians there are several types of the spontaneous abortion such as threatened, inevitable, incomplete, missed, septic, recurrent spontaneous, and complete abortion. Although for clinical purpose there are so many types of spontaneous abortion the pathology might be less different. Our study pointing out on the large, but understudied, group of missed abortions patients that would benefit from early diagnostic and preventive measures. Data about possible biomarkers as effective predictors of missed abortion are still scanty and the role of the immune system is not clear. We believe that such data as ours coming from MA patients are important to elucidate the role of the immune system and thus, a need or no for an immunomodulatory pre-treatment of MA patients going for next pregnancy.

To be more clear for the significance of the study we added a text in  the Introduction part (Lines 79-81): "The role of the immune system in MA is not clear. Data about possible biomarkers as as effective predictors of MA are still scanty" and one sentence in the Discussion part (lines 345-347): "We believe that our data regarding the understudied group of MA patients are important to elucidate the role of the immune system and in accumulation of data about (un)specific biomarkers for this condition".

All corrections are given in red in the revised version of the manuscript uploaded.